# Observed Survival Interval: A Supplement to TCGA Pan-Cancer Clinical Data Resource

**DOI:** 10.3390/cancers11030280

**Published:** 2019-02-26

**Authors:** Jie Xiong, Zhitong Bing, Shengyu Guo

**Affiliations:** 1Department of Epidemiology and Health Statistics, XiangYa School of Public Health, Central South University, Changsha 410078, China; xiongjie86@126.com; 2Department of Computational Physics, Institute of Modern Physics of Chinese Academy of Sciences, Lanzhou 730000, China; bingzt@impcas.ac.cn; 3Department of Public Management, College of Economic Management, Changsha University, Changsha 410022, China

**Keywords:** overall survival, observed survival interval, skin cutaneous melanoma, The Cancer Genome Atlas, omics

## Abstract

To drive high-quality omics translational research using The Cancer Genome Atlas (TCGA) data, a TCGA Pan-Cancer Clinical Data Resource was proposed. However, there is an out-of-step issue between clinical outcomes and the omics data of TCGA for skin cutaneous melanoma (SKCM), due to the majority of metastatic samples. In clinical cases, the survival time started from the initial SKCM diagnosis, while the omics data were characterized at TCGA sampling. This study aimed to address this issue by proposing an observed survival interval (OBS), which was defined as the time interval from TCGA sampling to patient death or last follow-up. We compared the OBS with the usual recommended overall survival (OS) by associating them with both clinical data and microRNA sequencing data of TCGA-SKCM. We found that the OS of primary SKCM was significantly shorter than that of metastatic SKCM, while the opposite happened if OBS was compared. OS was associated with the pathological stage of both primary and metastatic SKCM, while OBS was associated with the pathological stage of primary SKCM but not that of metastatic SKCM. Five previously cross-validated survival-associated microRNAs were found to be associated with the OBS rather than OS in metastatic SKCM. Thus, the OBS was more appropriate for associating microRNA-omics data of TCGA-SKCM than OS, and it is a timely supplement to TCGA Pan-Cancer Clinical Data Resource.

## 1. Introduction

Skin cutaneous melanoma (SKCM) is the most common malignant skin cancer and its incidence, mortality, and disease burden have been increasing annually [1,2,3]. Clinically, the American Joint Committee on Cancer (AJCC) staging is now the dominant synthetical index to predict SKCM prognosis [4]. Although useful, on the one hand, significant variability of prognosis in SKCM patients with the same AJCC pathological stage is observed [5]. On the other hand, it is hard to understand the underlying biology of SKCM just based on clinicopathological characteristics, and, further, it is difficult to apply individualized treatment protocols to SKCM patients [6].

By comprehensively characterizing molecular patterns in hundreds of SKCM samples, The Cancer Genome Atlas (TCGA) project has provided a comprehensive way to understand SKCM [7]. Multi-omics data with large sample sizes make the discovery of novel biomarkers that may potentially affect diagnosis, treatment and prognosis of SKCM possible. Several studies have been conducted to identify prognostic biomarkers based on various TCGA-SKCM omics data. Jayawardana et al. and Guo et al. proposed fifteen and five prognostic microRNAs (miRNAs) by mining TCGA-SKCM miRNA sequencing data, respectively [8,9]. Chen et al., Yang et al. and Ma et al. identified, from TCGA-SKCM RNA sequencing data, four, six, and six long non-coding RNAs for SKCM prognosis, respectively [10,11,12]. Furthermore, Jiang et al. focused on a multi-omics analysis by integration of mutation, copy number variation, methylation, and messenger RNA expression data to achieve this objective [13].

Methodologically, survival analysis (non-parametric methods, such as the Kaplan-Meier method, or the semi-parametric methods, such as Cox regression analysis) is now the dominant method to explore associations between outcome variables and the possible affecting factors. Therefore, the first step of survival analysis was to select an appropriate outcome variable. All previous studies adopted overall survival (OS), defined as the time interval from initial SKCM diagnosis to patient death or last follow-up [14], as the outcome variable. To ensure proper use of the large clinical dataset associated with omics features for TCGA users and to drive high quality survival outcome analytics, Liu et al. proposed an integrated TCGA Pan-Cancer Clinical Data Resource (TCGA-CDR), which includes four clinical outcomes and a list of outcome usage recommendations for each cancer type [14]. For SKCM, TCGA-CDR also recommends the use of OS for large-scale translational research. However, an out-of-step issue (or discordance) between OS and TCGA-SKCM omics data should be noticed. Specifically, TCGA didn’t always take the initially diagnosed SKCM samples for sequencing. Instead, SKCM samples from relapses or metastases in the follow-up of SKCM patients were usually adopted (i.e., SKCM samples submitted to TCGA were usually not the samples used for initial SKCM diagnosis). Therefore, omics data measured from TCGA-SKCM samples were out-of-step with the start time point of OS. This discordance will lead to biologically meaningless associations between OS and the omics data of TCGA-SKCM. Furthermore, prognostic biomarkers identified based on these associations will further misguide downstream experimental directions.

In this study, we aimed to address the out-of-step issue by proposing an observed survival interval and comparing it with OS in TCGA-SKCM dataset. Our findings prompted TCGA users to carefully select clinical outcomes when using TCGA-SKCM data for omics translational research.

## 2. Materials and Methods

### 2.1. Data Retrieval and Preprocessing

Level 1 clinical data, level 3 miRNA isoform sequencing raw counts, and the corresponding meta-data of SKCM samples were retrieved and downloaded from TCGA repository (https://cancergenome.nih.gov/). The retrieval strategies, exclusion criteria, preprocessing of miRNA sequencing data and clinical data [15] are presented in Appendix A, respectively. Hierarchical clustering was performed to detect sample outliers and guided principal component analysis (Appendix A) was used to evaluate the batch effects of the normalized miRNA isoform expression matrix [16].

### 2.2. Differential Expression Analysis

There were both primary SKCM (PCM) and metastatic SKCM (MCM) samples in TCGA-SKCM cohort. PCM samples with pathological stage I or II were defined as localized PCM (LPCM) samples and those with pathological stage III or IV were considered advanced PCM (APCM) samples [5]. Differential expression analysis was carried out to evaluate differences among LPCM, APCM, and MCM.MicroRNAs with at least a two-fold change of expression were considered to be biologically meaningful. The results of this analysis determined whether we would combine LPCM and APCM samples (i.e., PCM) or further combine PCM and MCM samples (i.e., SKCM) to identify prognostic miRNAs.

### 2.3. Observed Survival Interval

We defined an observed survival interval (OBS) as the time interval from TCGA sampling to patient death or last follow-up (Figure 1A). Unlike the usually adopted OS, the OBS has the same end time point as OS but different start time points, i.e., TCGA sampling introduced randomization to the start time point of OS. We obtained the DTS (days from initial SKCM diagnosis to TCGA sampling) and INPTS (indicator of new tumor event prior to TCGA sampling) from the parsed clinical files entitled “clinical_data.csv” and “new_tumor.csv”, respectively (see Appendix A for parsing clinical files). TCGA may take samples at different time points of SKCM progression. If TCGA took samples at initial SKCM diagnosis (i.e., DTS = 0), the OBS was equal to OS. If TCGA took samples at the first SKCM relapse or metastasis (i.e., DTS > 0 and INPTS = No), the OBS was equal to SAR (survival after the first relapse or metastasis). DTS was equal to PFI (progression-free interval) (Figure 1A) if TCGA took samples at subsequent SKCM relapses or metastases (i.e., the second/third/… relapse or metastasis). In practice, we could obtain the OBS by subtracting DTS from OS. For example, patient TCGA-W3-A825 survived for 1917 days from her initial SKCM diagnosis to death (i.e., OS = 1917). Furthermore, a MCM was found in her lung at 1644 days after initial SKCM diagnosis. TCGA didn’t obtain her initially diagnosed SKCM sample and therefore the MCM sample was taken for sequencing (i.e., DTS = 998). Thus, her OBS = OS − DTS = 273 days (Figure 1B).

### 2.4. Comparison of OS and OBS in Association with Clinical Data

The Kaplan-Meier survival analysis and log-rank test were applied to evaluate the prognostic effects of demographic and clinicopathological characteristics by considering both OS and OBS as clinical outcomes. The multivariate Cox regression model [17] was used to evaluate the independence of demographic and clinicopathological characteristics.

We also inferred the pathological stage at the time of TCGA sampling for MCM patients based on the 8th edition of the AJCC melanoma staging system [4]. Specifically, if the SKCM patient was initially diagnosed as pathological stage IV, the patient was still stage IV at the time of TCGA sampling; otherwise, if TCGA took a non-distant MCM sample from the patient, the patient was stage III at the time of TCGA sampling; otherwise, TCGA took a distant MCM sample from the patient and the patient should have been stage IV at the time of TCGA sampling. The prognostic effect of the inferred pathological stage was also evaluated by the Kaplan-Meier survival analysis.

### 2.5. Comparison of OS and OBS in Associating miRNA Sequencing Data

Univariate Cox regression analyses and proportional hazards assumption tests [18] were used to preliminarily explore the associations between OS or OBS and miRNA expression profiles. The Benjamini-Hochberg method was adopted for multiple testing corrections [19]. A stepwise multivariate Cox regression analysis with all preliminarily survival associated miRNAs as covariates was applied to construct an independent miRNA expression signature for SKCM prognosis.

The survival risk score (SRS), defined as the standard form of the prognostic index, was used as the synthetical index to represent the prognostic miRNA expression signature. The prognostic index was defined as a linear combination of the miRNA expression values weighted by the regression coefficients. Specifically,
(1)$$\mathrm{SRS}=\frac{\mathrm{PI}-\mathrm{mean}(\mathrm{PI})}{\mathrm{sd}(\mathrm{PI})}$$
where PI is a prognostic index vector and the *j*th element of PI is the prognostic index of the *j*th patient, i.e.,
(2)$${\mathrm{PI}}_{j}=\underset{i}{\sum }\beta_{i}\times E_{\mathrm{ij}}$$
where, *β_i_* is the multivariate Cox regression coefficient of the *i*th miRNA and *E_ij_* is the expression value of the *i*th miRNA in the *j*th sample. The ability of SRS to predict the SKCM patient survival outcome was assessed by calculating the area under the curve (AUC) of the time dependent receiver operating characteristic (ROC) at 3 years, 5 years, and 10 years, respectively [20].

### 2.6. Statistical Analysis

All analyses were done using R 3.4.4 [21]. MicroRNA sequencing raw counts normalization and differential expression analyses were conducted by a “DESeq2” package [22]. Guided principal component analysis was implemented by a “gPCA” package [16]. Survival analysis and proportional hazards assumption tests were performed by a “survival” package and a “survminer” package, respectively. Time dependent ROC analyses were done using a “timeROC” package [20]. All *P* values or adjusted *p*-values less than 0.05 were considered to be significant.

## 3. Results

### 3.1. TCGA-SKCM Dataset

There were 470 SKCM patients who provided 452 samples to TCGA for miRNA sequencing. Of the 452 SKCM samples, 97 were primary SKCM (PCM) samples, 352 were metastatic SKCM (MCM) samples, one was an additional MCM sample, and two were normal samples. We only analyzed the PCM and MCM samples due to the small number of normal and additional MCM samples. After preprocessing, 357 SKCM samples (82 PCM samples and 275 MCM samples) and 564 miRNAs were retained. To reproduce our analysis, the preprocessed clinical data and normalized miRNA sequencing data are presented in Datasets S1 and S2, respectively. Batch effect analyses of the normalized miRNA expression matrix showed that there was no discernible separation on the first two guided principal components (Appendix AA) with a permutation test *p*-value of 0.472 (Appendix AB). These results revealed that although TCGA-SKCM samples were sequenced in different batches, there was no significant batch effect among them.

### 3.2. Differences between OBS and OS

Of the 357 patients in TCGA-SKCM cohort, 171 patients were deceased and 186 patients were alive at the time of last follow-up. The median OS time and OBS time of TCGA-SKCM cohort were 2184 days (95% CI, 1927–3266 days) and 986 days (95% CI, 854–1276 days), respectively. For TCGA-PCM cohort, 18 patients were deceased and 64 patients were alive at the time of the last follow-up. The median OS time and OBS time of TCGA-PCM cohort were 1070 days (95% CI, 857–NA days) and 1276 days (95% CI, 1070–NA days), respectively. For TCGA-MCM cohort, 122 patients were deceased and 153 patients were alive at the time of last follow-up. The median OS time and OBS time of TCGA-MCM cohort were 2402 days (95% CI, 1992–3424 days) and 896 days (95% CI, 732–1175 days), respectively.

There was no obvious difference between OS and the OBS in TCGA-PCM cohort (*p* = 0.85) because the majority of the samples (80.49%) submitted to TCGA in this cohort were samples that were initially SKCM diagnosed. However, the OBS was significantly shorter than OS in TCGA-MCM cohort (*p* = 2.51×10^−11^) as the majority of the samples (90.55%) submitted to the TCGA in this cohort were not samples that were initially SKCM diagnosed, but samples excised from follow-up metastases (an average of a 1403 day delay after initial SKCM diagnosis). Furthermore, the median OS of TCGA-PCM cohort was significantly shorter than that of TCGA-MCM cohort (Figure 2A) and the opposite was true when the OBS was compared (Figure 2B). Logically, PCM patients were expected to survive longer than MCM patients. According to our analyses, the results were hard to explain if OS was adopted as the survival outcome, while it became explicable by considering the OBS as the survival outcome. These results revealed that OS and the OBS were different and the difference may give rise to distinct associations when used as survival outcomes in omics translational research.

### 3.3. OS Deemed More Appropriate to Associate Clinicopathological Characteristics than the OBS

Demographic and clinicopathological characteristics of TCGA-SKCM cohort measured at initial SKCM diagnosis are summarized in Table 1. The age at initial diagnosis, AJCC pathological stage, ulceration, and Breslow depth were significantly associated with OS of SKCM patients (Table 1). However, none of them were associated with the OBS (Appendix A). Furthermore, only the AJCC pathological stage was an independent predictor of OS of SKCM patients (Table 1).

Subgroup analysis revealed that SKCM patients with a higher pathological stage had shorter OS in bothTCGA-PCM cohort (HR = 3.63, 95%CI: 1.26–10.42; Appendix AA) and TCGA-MCM cohort (HR = 1.77, 95%CI: 1.25–2.52; Appendix AB). However, for the OBS, it was associated with the pathological stage in TCGA-PCM cohort (HR=3.63, 95%CI: 1.26–10.41; Appendix AC) but not in TCGA-MMC cohort (HR = 0.94, 95%CI: 0.67–1.30; Appendix AD).

As the AJCC pathological stage was the only independent predictor of OS of SKCM patients, we further inferred the pathological stage at the time of TCGA sampling for MCM patients. The inferred pathological stage was significantly associated with the OBS (HR = 2.06, 95%CI: 1.24–3.41; Figure 3B) rather than OS (HR = 1.03, 95%CI: 0.69–1.53; Figure 3A) in TCGA-MCM cohort.

The clinicopathological characteristics provided by TCGA were measured at the time of the initial SKCM diagnosis. Thus, they were in accordance with the start time point of OS, while they were usually out-of-step with respect to the OBS (especially for TCGA-MCM cohort). For TCGA-PCM cohort, OS and the OBS were usually the same, and indiscriminate usage of them will not result in a significant difference. Overall, these results indicated that the clinicopathological characteristics reasonably predicted OS, while they were not appropriate to predict the OBS due to the out-of-step issue.

### 3.4. Differentially Expressed miRNAs

Differentially expressed miRNAs between PCM and MCM were mainly in the hsa-miR-205-5p, hsa-miR-203a-3p, and hsa-miR-200 family (Figure 4). Only hsa-miR-3150b-3p was differentially expressed in APCM (advanced PCM, PCM with pathological stage III and IV) versus LPCM (localized PCM, PCM with pathological stage I and II), while it didn’t show any difference in APCM versus MCM or LPCM versus MCM (Figure 4). These results were consistent with the discoveries of Xu et al. on differentially expressed miRNAs in MCM versus PCM [23]. Furthermore, these results revealed that biological differences were mainly between PCM and MCM rather than between LPCM and APCM. Thus, it was appropriate to combine LPCM samples and APCM samples as PCM samples to explore survival associated miRNAs, while it was improper to further combine PCM samples and MCM samples as SKCM samples to identify survival associated miRNAs.

### 3.5. OBS Deemed More Appropriate to Associate miRNA-Omics Data than OS

Segura et al., Caramuta et al. and Tembe et al. proposed several miRNA expression signatures for MCM prognosis based on low sample size microarray data [24,25,26]. Jayawardana et al. [8] proposed fifteen prognostic miRNAs for stage III MCMs from TCGA miRNA-omics data and, further, systematically cross-validated these fifteen miRNAs with previous studies [24,25,26]. Five miRNAs (hsa-miR-142-5p, hsa-miR-150-5p, hsa-miR-342-3p, hsa-miR-155-5p, and hsa-miR-146b-5p) were found to be cross-validated (i.e., with greater validation rates across studies). However, none of the fifteen miRNAs identified from TCGA data overlapped with the five cross-validated miRNAs. Thus, Jayawardana et al. claimed that TCGA-MCM data performed the worst. These results were considered priori criteria for comparison of OS and the OBS in miRNA sequencing data.

Univariate Cox regression analysis revealed that there was no miRNA associated with either OS or the OBS in TCGA-PCM cohort. Meanwhile, 27 and nine miRNAs were found to be significantly associated with the OBS and OS of patients in TCGA-MCM cohort, respectively (Appendix A). Interestingly, all of the five cross-validated miRNAs were found to be associated with the OBS rather than OS in TCGA-MCM cohort (Appendix A) despite all being missed in the analysis of Jayawardana et al. due to the adoption of OS [8].

Unlike the clinicopathological characteristics measured at the time of initial SKCM diagnosis, the molecular patterns were characterized at the time of TCGA sampling. Thus, the miRNA-omics data were in accordance with the start time point of the OBS, while they were out-of-step with respect to OS (especially for TCGA-MCM cohort). Combined with the results from the above clinical analyses, OS and the OBS were usually the same in TCGA-PCM cohort, while they were usually different in TCGA-MCM cohort. Thus, for TCGA-MCM cohort, the OBS should be used for associating miRNA-omics data, and indiscriminate usage of OS and the OBS in TCGA-PCM cohort will not result in a significant difference. Overall, the OBS was more appropriate for identification of prognostic biomarkers than OS. Furthermore, to evaluate the independence and compare the prognostic power of clinicopathological characteristics and biomarkers, the time point of clinical data and omics data must be in accordance. As the time point of the omics data was fixed at TCGA sampling, it is wise to deduce the clinicopathological characteristics at the time point of TCGA sampling.

### 3.6. A miRNA Expression Signature for MCM Prognosis Based on the OBS

Although Jayawardana et al. have cross-validated five miRNAs for MCM prognosis, the cross-validation was based on differential expression analysis between MCM patients with longer OS and MCM patients with poor OS [8]. Thus, relationships among the five miRNAs were not investigated further. For the identification of prognostic biomarkers, the construction of signatures that included as many independent biomarkers as possible was expected [15]. Correlation analysis showed that the five cross-validated miRNAs were correlated with each other (Figure 5A) and a collinearity existed among them (Kappa value = 70.32). Furthermore, multivariate Cox regression analyses revealed that hsa-miR-155-5p was an appropriate representative for the five cross-validated miRNAs (Appendix A).

Stepwise multivariate Cox regression analyses, by considering the OBS to be the survival outcome and the 27 OBS associated miRNAs as covariates, revealed that hsa-miR-155-5p, hsa-miR-4461, hsa-miR-504-5p, hsa-miR-625-5p, and hsa-miR-664b-5p were independent prognostic miRNAs for patients with MCM (Table 2, upper part). Model diagnosis revealed that the final regression model (Global Schoenfeld test *p* = 0.49) and all covariates in the model satisfied the proportional hazards assumption (Appendix A) with negligible collinearity (Kappa value = 2.70).

The survival risk score (SRS) was calculated for each MCM patient and Kaplan-Meier survival analysis showed that high-risk (SRS>0) MCM patients had shorter OBS compared with low-risk (SRS ≤ 0) MCM patients (HR = 3.29; 95%CI: 2.37–4.56; Figure 5B). Furthermore, as a continuous variable, the SRS was inversely associated with the OBS (HR = 2.32; 95%CI: 1.93–2.78; Wald test *p*< 0.001). The area under the curve of the receiver operating characteristic for SRS were 0.77 (95%CI: 0.71–0.84), 0.76 (95%CI: 0.68–0.83), and 0.79 (95%CI: 0.71–0.88) at three years, five years, and 10 years, respectively (Figure 5C). Finally, multivariate Cox regression analysis showed that the SRS was an independent predictor of the OBS of MCM patients while the inferred pathological stage was no longer significant (Table 2, lower part).

## 4. Discussion

The recently proposed TCGA-CDR includes OS, PFI (progression-free interval), DFI (disease-free survival), DSS (disease-specific survival), and a list of outcome usage recommendations for each cancer type [14]. As indicated by Liu et al., TCGA mainly collected primary tumors for molecular characterization, with the exception of the SKCM study, which included mainly metastatic samples. According to our results, there was no statistically significant difference between OS and the OBS in PCM (*p* = 0.85). Thus, TCGA-CDR prudently recommended using only the limited number of PCM samples for SKCM clinical outcome correlations [14]. However, this recommendation discarded the majority of the MCM samples. Although there was no statistically significant difference between OS and the OBS in TCGA-PCM cohort, OS and the OBS were not always the same for every PCM patient (19.51%). This means that some of the PCM samples submitted to TCGA may not be samples obtained at initial SKCM diagnosis but relapse samples from the follow-up. For example, patient TCGA-ER-A2NB provided a PCM sample to TCGA, but his DTS = 124 days. Although we believe this non-significant difference between OS and the OBS will not lead to significantly different associations when applying to TCGA-PCM omics data, a more precise recommendation is necessary.

Some limitations of the current study should be noticed. (I) We addressed the out-of-step issue of survival outcome but various outcome endpoints based on relapses or metastases were not investigated. With the exception of OS, Liu et al. also recommended PFI and DSS for TCGA-SKCM data. However, there also existed the out-of-step issue for these outcomes in MCM. For example, all outcome endpoints of TCGA-W3-A825 in TCGA-CDR were 1917 days [14]. According to our analyses, the PFI of TCGA-W3-A825 should be 1644 days (i.e., DTS = 1644 days; Figure 1B), but this outcome was not appropriate for the investigation of associations between it and omics data due to no primary sample being available for TCGA-W3-A825. (II) We didn’t find any prognostic miRNAs to predict the survival of PCM. Possible explanations are: (a) PCM is a relatively early event in SKCM progression and PCMomics data may be unable to predict the relatively long survival-based outcomes such as OS and the OBS. Thus, relatively shorter outcome endpoints based on relapses or metastases may be appropriate for PCMomics translational research. (b) Single miRNA-omics may not be enough for predicting PCM patient survival, and a multi-omics analysis should be conducted. (III) Despite this study focusing on the outcome endpoints of TCGA-SKCM data, the five identified novel prognostic miRNAs based on the OBS call for additional studies for further validation and mechanism exploration.

## 5. Conclusions

In conclusion, we defined the OBS, to supplement TCGA-CDR, and recommended it for TCGA-SKCMomics translational research. Our results could remind subsequent TCGA-SKCM data users to pay attention to the out-of-step issue of outcome endpoints. Although our analyses were based on associations between survival-based outcomes and TCGA-SKCM miRNA-omics data, they could be generalized to associate other TCGA-SKCMomics data and relapses, or metastasis-based outcomes. In addition, the five identified prognostic miRNAs may be of value in predicting the OBS of MCM patients and informing future experimental investigations.

## Figures and Tables

**Figure 1 cancers-11-00280-f001:**
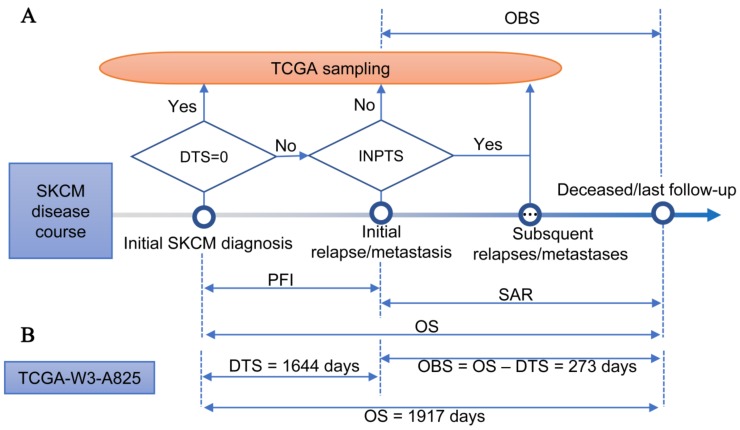
Definition of observed survival interval for TCGA-SKCM cohort. (**A**) Disease course of TCGA-SKCM cohort. DTS—days from initial SKCM diagnosis to TCGA sampling; INPTS—indicator of new tumor event prior to TCGA sampling; SAR—survival after the first relapse/metastasis; OS—overall survival; OBS—observed survival interval; PFI—progression-free interval. The diamond-shaped box denotes the examination of the condition included in the box. (**B**) Disease course of patient TCGA-W3-A825.

**Figure 2 cancers-11-00280-f002:**
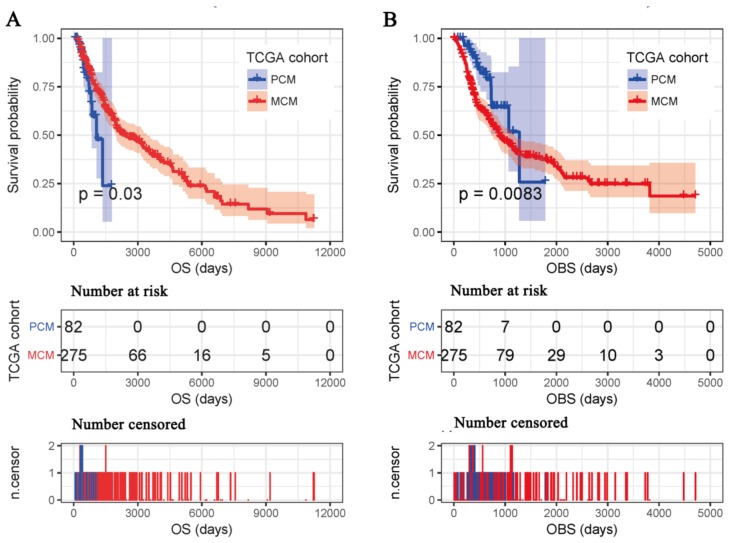
Kaplan-Meier survival analysis between PCM patients and MCM patients. PCM—primary SKCM; MCM—metastatic SKCM. OS (**A**) and the OBS (**B**) were considered clinical outcomes. The upper, middle, and lower parts represent the Kaplan-Meier plot, number of patients at risk for each group, and number of censored patients for each group respectively. OS—overall survival; OBS—observed survival interval.

**Figure 3 cancers-11-00280-f003:**
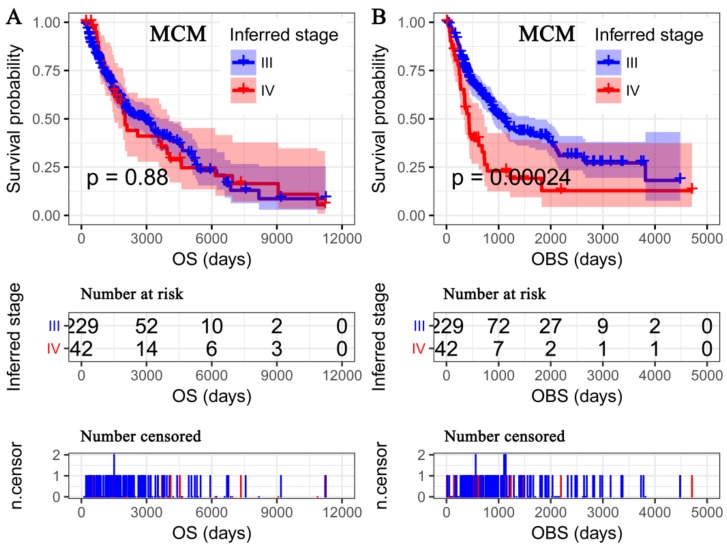
Kaplan-Meier survival analysis of inferred pathological stage. The OBS and inferred pathological stage in TCGA-MCM cohort (**A**), OS and inferred pathological stage in TCGA-MCM cohort (**B**). OS—overall survival; OBS—observed survival interval.

**Figure 4 cancers-11-00280-f004:**
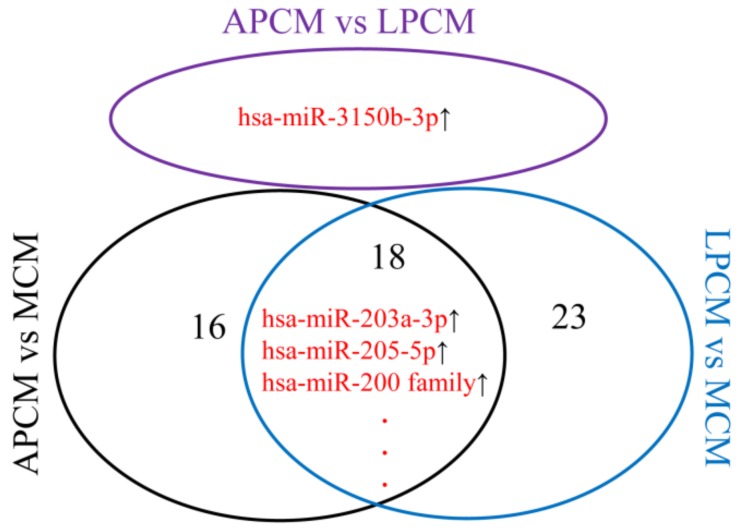
Venn plot of differentially expressed miRNAs of APCM versus LPCM, APCM versus MCM, and LPCM versus MCM. Arrow and number represent regulation direction and number of differentially expressed miRNAs, respectively. LPCM—localized primary SKCM; APCM—advanced primary SKCM; MCM—metastatic SKCM.

**Figure 5 cancers-11-00280-f005:**
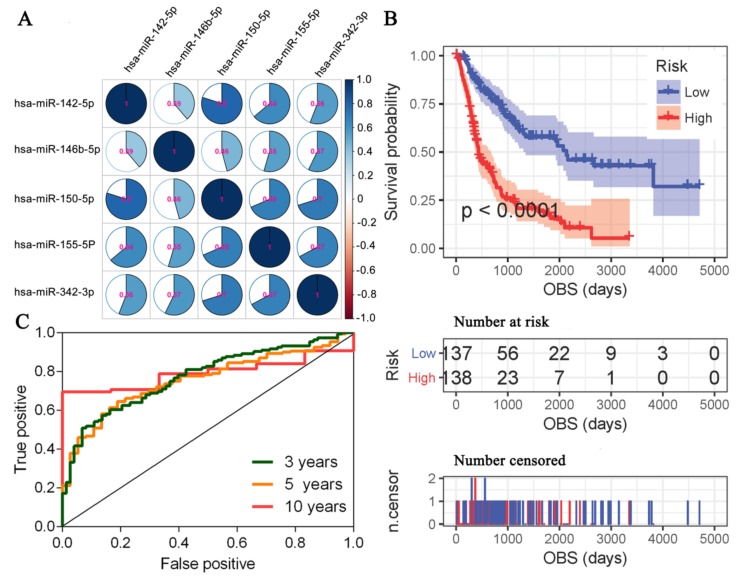
Prognostic miRNAs for MCM based on the OBS. (**A**) Pairwise Spearman correlations ofthe five cross-validated prognostic miRNAs in MCM. (**B**) Kaplan-Meier survival analysis between high-risk and low-risk MCM patients. (**C**) Time dependent receiver operating characteristic curves of SRS at different times. SRS-survival risk score.

**Table 1 cancers-11-00280-t001:** Survival analysis of demographic and clinicopathological characteristics based on OS.

Column Header	Deaths/Patients (%)	MS (95% CI)	^U^Log-rank test *p*	^M^HR (95% CI)	^M^Wald test *p*
**Age** ^1^					
≤50 years	58/113 (51.33)	4062 (2022–5370)			
>50 years	115/244 (47.13)	1927 (1524–2927)	0.011	1.27 (0.82–1.96)	0.282
**Gender**					
Male	60/138 (43.48)	2004 (1640–4507)			
Female	111/219 (50.68)	2402 (1960–3424)	0.736		
**Breslow depth** ^1^					
≤2 mm	55/114 (48.25)	3943 (3139–5318)			
>2 mm	80/169 (47.33)	1424 (1103–2004)	<0.001	1.44 (0.93–2.22)	0.098
**Pathological stage** ^2^					
I–II	81/179 (45.25)	3266 (2402–4601)			
III–IV	75/153 (49.02)	1490 (988–2071)	<0.001	1.82 (1.22–2.72)	0.004
**Ulceration** ^1^					
No	57/111 (51.35)	2402 (1927–4222)			
Yes	58/133 (43.61)	1354 (1059–2028)	<0.001	1.53 (1.00–2.35)	0.052
**Primary tumor site**					
Extremities	76/158 (48.1)	2071 (1910–4000)			
Head and neck	11/23 (47.83)	2192 (787–NA)			
Trunk	59/128 (46.09)	3139 (1691–5107)	0.787		
**Radiation therapy**					
No	165/355 (49.25)	2192 (1917–3266)			
Yes	7/14 (50.00)	1341–NA	0.892		
**Chemotherapy**					
No	121/251 (48.21)	2173 (1832–3564)			
Yes	40/75 (53.33)	2184 (1917–3683)	0.813		

MS—median survival; CI—confidence interval; HR—hazard ratio; ^U^—univariate analysis; ^M^—multivariate analysis. For the multivariate Cox regression analyses, variable coding are: age (1, ≤50 years; 2, >50 years), pathological stage (1, I–II; 2, III–IV), ulceration (1, No; 2, Yes), and Breslow depth (1, ≤2 mm; 2, >2 mm). ^1^Significant in univariate analysis; ^2^Significant in multivariate analysis. Patients with missing values were omitted from the table.

**Table 2 cancers-11-00280-t002:** Independent prognostic miRNAs for MCM patients based on OBS.

Column Header	^M^HR (95% CI)	^M^Wald Test *P*	Type
hsa-miR-155-5p	0.73 (0.63–0.85)	3.15×10^−5^	Protective^1^
hsa-miR-4461	1.29 (1.13–1.46)	1.07×10^−4^	Risky^2^
hsa-miR-504-5p	0.80 (0.71–0.92)	1.17×10^−3^	Protective
hsa-miR-625-5p	0.67 (0.53–0.86)	1.35×10^−3^	Protective
hsa-miR-664b-5p	0.69 (0.58–0.83)	4.39×10^−5^	Protective
SRS	2.28 (1.89–2.74)	<2.00×10^−16^	Risky
Inferred stage	1.32 (0.88–1.98)	0.18	Risky

^M^—multivariate analysis; ^1^—HR<1; ^2^—HR>1.

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
