# Peer review of "Observed Survival Interval: A Supplement to TCGA Pan-Cancer Clinical Data Resource"

_cancers, 2019, doi:10.3390/cancers11030280_

Reviewer 1 Report

An important study for any biomarker discovery or validation study aiming at using TCGA data in melanoma. In addition to propose an appropriate and recomputed survival measure, the study proposes a redefined pathological stage.

Minor comments:

In the PCA looking for a batch effect, the permutation test is not described in material and methods. The authors mention "the first two guided components", it is unclear how these components are "guided" or if it is simply a misguiding use of the word.

It could be made more clear in which case OS (diagnostic-association clinicopathology and demography) and in which case OBS (TCGA sample related variables, eg omics) should be used. 

Minor English corrections:

224: biological differences were mainly existed between PCM and MCM 

240:  These results were considered as priori standards

247: due to the OS was adopted 

I am not sure that "out-of-step" is used correctly in this case.

Author Response

Response to Reviewer 1 Comments

Point 1: An important study for any biomarker discovery or validation study aiming at using TCGA data in melanoma. In addition to propose an appropriate and recomputed survival measure, the study proposes a redefined pathological stage.

Response 1: Thank you for these positive summarizations. The reviewer has exactly summarized our work.

Point 2: In the PCA looking for a batch effect, the permutation test is not described in material and methods. The authors mention "the first two guided components", it is unclear how these components are "guided" or if it is simply a misguiding use of the word.

Response 2: Thank you for this comment. Guided PCA (gPCA) is an extension of traditional PCA by incorporating batch information into the expression matrix (i.e., batch indicator matrix multiple expression matrix). Instead of performing Singular-value decomposition (SVD) on the expression matrix in traditional PCA, gPCA performing SVD on the batch indicator transformed expression matrix. Large singular values imply that the batch is important for the corresponding principal component. Thus, gPCA guides the SVD to look for batch effects in the data based on the batch indicator matrix. Furthermore, compared to subjective visual inspection of the first and second principal components in traditional PCA, gPCA proposed a statistic (i.e., the ratio of the variance of the first principal component from gPCA to the variance of the first principal component from traditional PCA) for permutation test. A permutation distribution is created by permuting the batch vector 1000 times so that the test statistic is computed for each permutation. A one-sided P-value is estimated as the proportion of times the observed statistic value was in the extreme tail of the permutation distribution.

Because gPCA is proposed by Resse et al. [ref. 16], we didn’t detail the derivation of gPCA. In this revision, we added the above summarizations of gPCA in Section V of the Supplementary Materials.

Point 3: It could be made more clear in which case OS (diagnostic-association clinicopathology and demography) and in which case OBS (TCGA sample related variables, eg omics) should be used.

Response 3: Thank you for this comment. We have made the usage of OS (Line 237-241) and OBS (Line 285-293) more clear in this revision.

Point 4: Minor English corrections

224: biological differences were mainly existed between PCM and MCM

240: These results were considered as priori standards

247: due to the OS was adopted

Response 4: Thank you for these comments. These English corrections have been corrected in this revision. Specifically,

“biological differences were mainly existed between PCM and MCM” is corrected by “biological differences were mainly between PCM and MCM” (Line 250).

“These results were considered as priori standards” is corrected by “These results were considered as priori criteria” (Line 274).

“due to the OS was adopted” is corrected by “due to the adoption of OS” (Line 281).

We also checked the manuscript for possible grammar and spell errors.

Point 5: I am not sure that "out-of-step" is used correctly in this case.

Response 5: Thank you for this comment. We have illustrated the meaning of out-of-step issue in the manuscript (Line 64-68). In this revision, we also annotated “out-of-step” by “discordance” (Line 63).

Reviewer 2 Report

Major comments

I would like to congratulate Xiong et al. with the research they have done and the manuscript they have produced. Their work describes the limitations of overall survival (OS) for the evaluation of prognostic genomic biomarkers, specifically in TCGA data for skin cutaneous melanoma, caused by the so-called “out-of-step” issue, in which sampling (and genomic analyses) happened much later than the initial clinical diagnosis. To resolve this problem, the authors propose to use the observed survival (OBS), which measures the time between the moment a sample was taken and death or the end of followup, instead of OS. They further showed that OBS was more associated with the expression of published prognostic miRNAs than OS. These results will be of interest to anyone looking to use the TCGA melanoma data for prognostic analysis.

This being said, I do have a few questions and comments for the authors, the first of which is what the relevance could be of this work outside of the (relatively limited) use case described in the manuscript? Is the out-of-step issue something that has been previously described or that appears in other studies or perhaps clinical trials?

The authors claim that OBS is better suited for the survival analysis of omics data and provide some support for this claim with their analysis of the survival-associated miRNAs. This validation is however quite limited in scope and before they can claim better association of OBS with omics data in general, the authors should provide evidence from other types of omics data (or narrow down their claim). The TCGA database contains many different types of omics data, such as proteomics, gene expression, somatic mutations and DNA methylation data. And a cursory PubMed search brought up many different types of prognostic biomarkers for melanoma, so it should be relatively straightforward to test the improved association between OBS and omics data on different types of genomic prognostic markers outside of the five miRNAs described in the manuscript.

Minor comments

I would recommend switching the OS and OBS figures in figure 3 so that they are in the same order as figure 2 (OS left and OBS right). This will help avoid any possible confusion.

On line 54 and 55, the authors write that:

Recently, Liu et al. have demonstrated the validity and utility of TCGA data for cancer omics translational research [14].

However, the Liu et al. paper only focuses on the usage of TCGA data for survival analytics and the validity and utility of TCGA data for translational research has already been extensively described in earlier publications of the TCGA research network (see https://cancergenome.nih.gov/publications for a complete overview). I would therefore suggest removing the reference to the Liu et al. paper here.

In the discussion, the authors explain their motivation for using the TCGA SKCM data, namely the fact that this is the only TCGA study that included many metastatic samples. I think that it would be helpful to already mention this in the abstract, because it will help the reader understand the motivation and context of this research from the start. In addition, I think the abstract could also use a (short!) explanation of the term “out-of-step issue”, which many readers might not be familiar with.

The discussion of the results of Jayawardana et al. from line 234 to 241 is confusing. According to the authors, Jayawardana et al. cross-validated their fifteen prognostic miRNAs derived from TCGA data with previously published prognostic miRNAs and found that five miRNAs could indeed be cross-validated. This statement is then immediately followed by another statement (line 239) that says that none of the TCGA-derived miRNAs were cross-validated. Either the TCGA-derived miRNAs were validated or they were not, but the way this part is currently formulated, it is hard to tell.

The authors are probably already aware of this, but the NIH already has a definition for observed survival (https://surveillance.cancer.gov/survival/measures.html):

Observed survival is an estimate of the probability of surviving all causes of death.

I just wanted to point out the potential confusion that might result from using an existing term to name a different concept.

Author Response

Response to Reviewer 2 Comments

Point 1: I would like to congratulate Xiong et al. with the research they have done and the manuscript they have produced. Their work describes the limitations of overall survival (OS) for the evaluation of prognostic genomic biomarkers, specifically in TCGA data for skin cutaneous melanoma, caused by the so-called “out-of-step” issue, in which sampling (and genomic analyses) happened much later than the initial clinical diagnosis. To resolve this problem, the authors propose to use the observed survival (OBS), which measures the time between the moment a sample was taken and death or the end of followup, instead of OS. They further showed that OBS was more associated with the expression of published prognostic miRNAs than OS. These results will be of interest to anyone looking to use the TCGA melanoma data for prognostic analysis.

Response 1: Thank you for these positive summarizations. The reviewer has exactly summarized our work.

Point 2: This being said, I do have a few questions and comments for the authors, the first of which is what the relevance could be of this work outside of the (relatively limited) use case described in the manuscript? Is the out-of-step issue something that has been previously described or that appears in other studies or perhaps clinical trials?

Response 2: Thank you for this comment. The current study is aimed to provide a correct usage of TCGA-SKCM survival data to associate omics data. Except for directly use it to explore prognostic biomarkers, it can be used to associate any biological results derived from TCGA-SKCM data such as molecular subtypes.

We speculate that Liu et al [ref. 14] may have noticed this problem for TCGA-SKCM. However, they didn’t describe and solve this problem. As described in the discussion of our manuscript, Liu et al. prudently recommended using only the limited number of PCM samples for SKCM clinical outcome correlations. Because there was no statistical significant difference between OS and OBS in TCGA-PCM cohort, their recommendation will not result in significant difference. However, their recommendation discarded the majority MCM samples. In clinical trials, objects were usually sampled at initially diagnosis and variables were usually measured at sampling. Thus, OS and OBS were usually in accordance. In this study, we just focus on the TCGA-SKCM data due to the importance of TCGA data for cancer genome research.

Point 3: The authors claim that OBS is better suited for the survival analysis of omics data and provide some support for this claim with their analysis of the survival-associated miRNAs. This validation is however quite limited in scope and before they can claim better association of OBS with omics data in general, the authors should provide evidence from other types of omics data (or narrow down their claim). The TCGA database contains many different types of omics data, such as proteomics, gene expression, somatic mutations and DNA methylation data. And a cursory PubMed search brought up many different types of prognostic biomarkers for melanoma, so it should be relatively straightforward to test the improved association between OBS and omics data on different types of genomic prognostic markers outside of the five miRNAs described in the manuscript.

Response 3: Thank you for this comment. As pointed by the reviewer, we just provide evidence from miRNA-omics data. The reason is that we need some already validated prognostic biomarkers as comparison criteria to compare OS and OBS in associating omics data. Jayawardana et al have provided five such cross-validated prognostic miRNAs [8]. Thus, we choose miRNA-omics to illustrate our findings. We also narrowed our claim in the abstract and results by “miRNA-omics” (Line 27, 265, 283, 288). Finally, we outlook the usage of OBS for other TCGA-SKCM omics data in the conclusion of the manuscript (Line 375-378).

Point 4: I would recommend switching the OS and OBS figures in figure 3 so that they are in the same order as figure 2 (OS left and OBS right). This will help avoid any possible confusion.

Response 4: Thank you for this comment. Figure 3 is switched to the same order as figure 2, and the citation order of Figure 3 is changed in the manuscript.

Point 5: On line 54 and 55, the authors write that:

Recently, Liu et al. have demonstrated the validity and utility of TCGA data for cancer omics translational research [14].

However, the Liu et al. paper only focuses on the usage of TCGA data for survival analytics and the validity and utility of TCGA data for translational research has already been extensively described in earlier publications of the TCGA research network (see https://cancergenome.nih.gov/publications for a complete overview). I would therefore suggest removing the reference to the Liu et al. paper here.

Response 5: Thank you for this comment. We have deleted this description in this revision.

Point 6: In the discussion, the authors explain their motivation for using the TCGA SKCM data, namely the fact that this is the only TCGA study that included many metastatic samples. I think that it would be helpful to already mention this in the abstract, because it will help the reader understand the motivation and context of this research from the start. In addition, I think the abstract could also use a (short!) explanation of the term “out-of-step issue”, which many readers might not be familiar with.

Response 6: Thank you for this comment. We have added the descriptions of metastatic SKCM sample issue and out-of-step issue in the abstract of this revision (Line 16-17).

Point 7: The discussion of the results of Jayawardana et al. from line 234 to 241 is confusing. According to the authors, Jayawardana et al. cross-validated their fifteen prognostic miRNAs derived from TCGA data with previously published prognostic miRNAs and found that five miRNAs could indeed be cross-validated. This statement is then immediately followed by another statement (line 239) that says that none of the TCGA-derived miRNAs were cross-validated. Either the TCGA-derived miRNAs were validated or they were not, but the way this part is currently formulated, it is hard to tell.

Response 7: Thank you for this comment. Sorry for this ambiguous description, the results of Jayawardana et al. [ref. 8] are important and of interest to this study. The main reason for this ambiguity is caused by “cross-validated”. In the manuscript, “cross-validated” didn’t mean that all five miRNAs were found in TCGA-derived miRNAs and every previous study [ref. 24-26]. Indeed, in the cross-validation study of Jayawardana et al., “cross-validated” was just mean larger validation rate across studies. The TCGA-derived miRNAs were not validated by Jayawardana et al.. Thus, they claimed that TCGA-SKCM performed the worst in their cross-validation. Our analysis revealed that the reason that TCGA-derived miRNAs could not be validated by Jayawardana et al. was that they used OS as survival outcome when analyzing the TCGA-SKCM data. We have modified our description in this revision by illustrating the meaning of “cross-validated” (Line 271).

Point 8: The authors are probably already aware of this, but the NIH already has a definition for observed survival (https://surveillance.cancer.gov/survival/measures.html):

Observed survival is an estimate of the probability of surviving all causes of death.

I just wanted to point out the potential confusion that might result from using an existing term to name a different concept.

Response 8: Thank you for this comment. As stated in Point 2, our definition is used for the TCGA-SKCM data. It will be of interest only to TCGA-SKCM data users. Furthermore, OBS is clearly defined in this study and “observed” is a vivid description. To avoid potential confusion, “observed survival” is modified by “observed survival interval”, but “OBS” is still used as abbreviation.

Round  2

Reviewer 2 Report

Dear authors,

Thank you for the corrections you made to your manuscript. It has certainly improved. Clearly defining the scope of the manuscript, instead of the original broad claims, will help readers assess what the relevance of your work is to theirs.

Once the remaining language mistakes are fixed, I believe the manuscript will be ready for publications.